# Tolerance to soil acidity of soybean (*Glycine max L*.) genotypes under field conditions Southwestern Ethiopia

**Tolossa Ameyu Bedassa**[1,2]*, **Abush Tesfaye Abebe**[3], **Alemayehu Regassa Tolessa**[4]

**1** Jimma Agricultural Research Center, Jimma, Ethiopia, **2** Ethiopian Institute of Agricultural Research, Addis Ababa, Ethiopia, **3** International Institute of Tropical Agriculture, Ibadan, Nigeria, **4** Department of Natural Resource Management, Jimma University, Jimma, Ethiopia

* tolamboo@gmail.com

**Data Availability Statement:** All relevant data are within the paper and its Supporting Information files.

## Abstract

Soil acidity with associated low nutrient availability is one of the major constraints to soybean production in southwestern Ethiopia. Integrated use of lime and acid-tolerant crops is believed to reduce soil acidity and improve crop production. The experiment was conducted in the field condition of Mettu, southwestern Ethiopia during the 2017/18 main cropping season. The experiment comprised fifteen soybean genotypes and two soil amendment (lime and unlimed) treatments arranged in a split-plot design with three replications. For each treatment, four rows were planted per plot; data related to growth, root, nodule, and yield of the crop were collected at a necessary stage for each. Liming and genotype interaction had significantly ($P = 0.01$) affected all parameters considered except for hundred seed weight and root volume and were affected only by the main effects of genotypes and liming. A significant reduction for most parameters was found on lime-untreated soil than treated soil. Though some genotypes showed higher performance for root, growth parameters, and yield components under unlimed soils; however, gave higher yield and yield components, when grown on lime-untreated with an average yield reduction of 13.7%, due to soil acidity. The maximum grain yield of (1943.93 kg ha$^{-1}$) was obtained under lime treated acid soil from PI567046A genotype; while the lowest (510.49 kg ha$^{-1}$) were recorded from SCS-1genotype under the lime untreated acid soil. Genotype BRS268 showed higher yield (1319.83 kg ha$^{-1}$) under lime untreated acid soil than lime treated acid soil (1143.47 kg ha$^{-1}$) and showed less reduction percentage for a number of the nodules, root weight, and number of seeds per plant; while PI567046A showed high reduction percentage for yield, biomass, number of pod and seed per plant. A high difference was observed among the soybean genotypes for soil acidity tolerance, which might be further exploited by breeders for the genetic improvement of soybean. Genotype BRS268 had performed better than other tested genotypes under increased soil acidity. selection would be effective to improve soybean genotypes performance on acid soils and identify low Phosphorus tolerant genotype that helps smallholder farmers optimize soybean productivity on acid soils in the study area. HAWASSA-04 variety is *the most tolerant among the tested materials. However, further study is required by considering additional genotypes to reach a conclusive recommendation.*

**Funding:** The funder of this paper was only the Ethiopian Institute of Agricultural Research and Authors only. As I indicated in the first draft of the submission, the Ethiopian Institute of Agricultural Research covers daily labor to conduct this activity, providing planting materials(seed), providing land to conduct activities, However, the Ethiopian Institute of Agricultural research has no role in data collection, analysis, preparation of manuscript but those authors have.

**Competing interests:** The authors have declared that no competing interests exist.

## Introduction

Land degradation, soil nutrient depletion and increasing soil acidity is challenging problem in south western Ethiopia. Soil acidity is one of the major problem that have profound effect on the productive potential of crops, such as soybean, because of low availability of basic cations, and excess and toxic levels of hydrogen and aluminum in exchangeable forms [1]. The major causes for soils to become acidic are high rainfall, leaching, acidic parent material, organic matter decay, and harvest of high yielding crops. Crop management practices (continuous application of acid forming fertilizers), removal of organic matter, and contact exchange between exchangeable hydrogen on root surfaces and microbial production of nitric and sulfuric acids can also contribute to soil acidity [2]. Soil acidity is often an insidious soil degradation process, developing slowly; although indicators, such as falling yields, leaf discolorations in susceptible plants, and lack of response to fertilizers might indicate that soil pH is declining to critical levels [3]. Theoretically, soil acidity is quantified based on $H^+$ and $Al^{3+}$ concentrations of the soils [4]. Acidic soils limit the production potential of crops because of low availability of basic cations and excess of hydrogen (H+) and aluminum ($Al^{3+)}$) in exchangeable forms. It affects beneficial microorganisms, reduced root growth, which limits absorption of nutrients and water [5], consequently, leading to poor plant growth and yield of crops. However, $Al^{3+}$ toxicity is one of the major limiting factors for crop production on acid soils by inhibiting root cell division and elongation, thereby, reducing water and nutrient uptake [6] poor nodulation or mycorrhizal infections.

Acidic soil is mostly distributed in developing countries, where there is high population growth, and food demand is ever increasing. Acid soils make up approximately 30% of the world's total land area and more than 50% of the world's potentially arable lands, particularly in the tropics and subtropics [7]. Acidic soils cover a total of 1.66 billion hectares in developing countries, while the total area affected by soil acidity is about 4 billion hectares [8]. In high rainfall areas, excessive rainfall coupled with unfavorable temperature and precipitation is high enough to leach appreciable amounts of exchangeable basic cations [9].

Soybean production and productivity have been growing rapidly in Ethiopia, in the past decade. According to the Agricultural Sample Survey of CSA (Central Statics Agency) [10], 130,022.00 private peasant holdings cultivated about 36,635.79 hectares of land and produced about 812, 34.659 tons of soybean. The average production of soybean in the country is, therefore, 2.2 t ha$^{-1}$ while, that of the Mettu area is by far below (1.3 t ha$^{-1}$) the national average due to soil acidity [11].

Lime and fertilizer management practices are of primary importance for the proper management of soil acidity. Application of lime significantly increased root and shoot yields of soybean in Nigeria [12]. Nevertheless, for economic reasons, it is often not practicable for resource-poor farmers to apply high rates of lime [13]. And [14] also reported that application of lime with high rate is not practicable for resource-poor farmers, as well as, mineral fertilizers. However, previous studies revealed the existence of sufficient genotypic variability of bean germplasm for acid-tolerant [15, 16]. Hence, identification of tolerance soybean genotypes to soil acidity is an economically feasible option that might serve as an acid soil management practice [11]. Hence, the identification and use of soybean genotypes that are tolerant to acid soil conditions of Southwestern Ethiopia is a very useful approach to ensure economic stability to many subsistence farmers, who cannot afford the application of liming materials practices [11]. A preliminary field screening of soybean genotypes in southwestern Ethiopia has demonstrated the presence of genetic variability among genotypes in tolerating soil acidity stress. Studying responses of selected genotypes with contrasting tolerance to soil acidity may help in generating information that could be utilized by breeding programs aimed at developing

aluminum-tolerant cultivars for areas where soil acidity remains a key environmental constraint to crop production. Therefore, to meet the demand of soybean in Ethiopia including the study area, emphasis should be given to increase the productivity of the crops through the use of genotypes that can tolerate acid stressed soil conditions. The objective of this study was to test the hypothesis that differences exist in growth, root, yield and yield parameters among soybean genotypes selected for soil acidity tolerance when subjected to limed and unlimed acid soil.

## Materials and methods

### Description of the study site

The field experiment was conducted at Mettu Agricultural Research Sub Center. Mettu is located in south western Ethiopia at 8˚19' 0" N latitude, 35˚35' 0"E longitude, and at the altitude of 1550 meters above sea level. The average annual rainfall of the study site was 1835 mm/annum, an annual mean minimum and maximum temperatures were 12 and 27 $^0$C respectively (Figs 1 and 2).

The soil of the study area has a pH (H2O) value of 4.5, exchangeable acidity of 2.82 cmol kg-1 soil and soil available phosphorus level of 1.16 ppm before applying the treatments. Physical and some chemical properties of the soil in the study area before sowing and after harvesting are presented (Table1).

### Soil Sampling and analysis

Prior to the field experimentation both undisturbed and disturbed samples were collected. Three undisturbed soil samples were taken by core sampler. Fresh weight and an oven dry

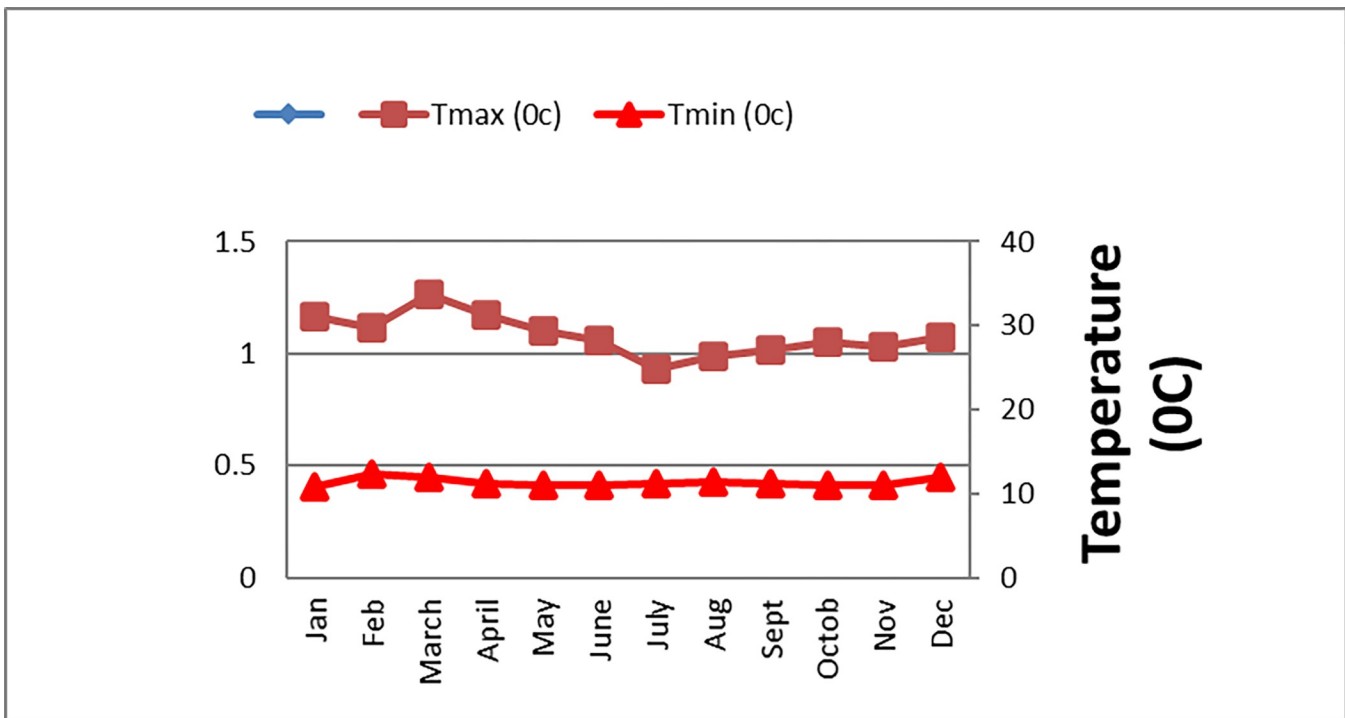

**Fig 1. Mean minimum and maximum temperatures (˚C) of Mettu during crop growth period in 2017.**

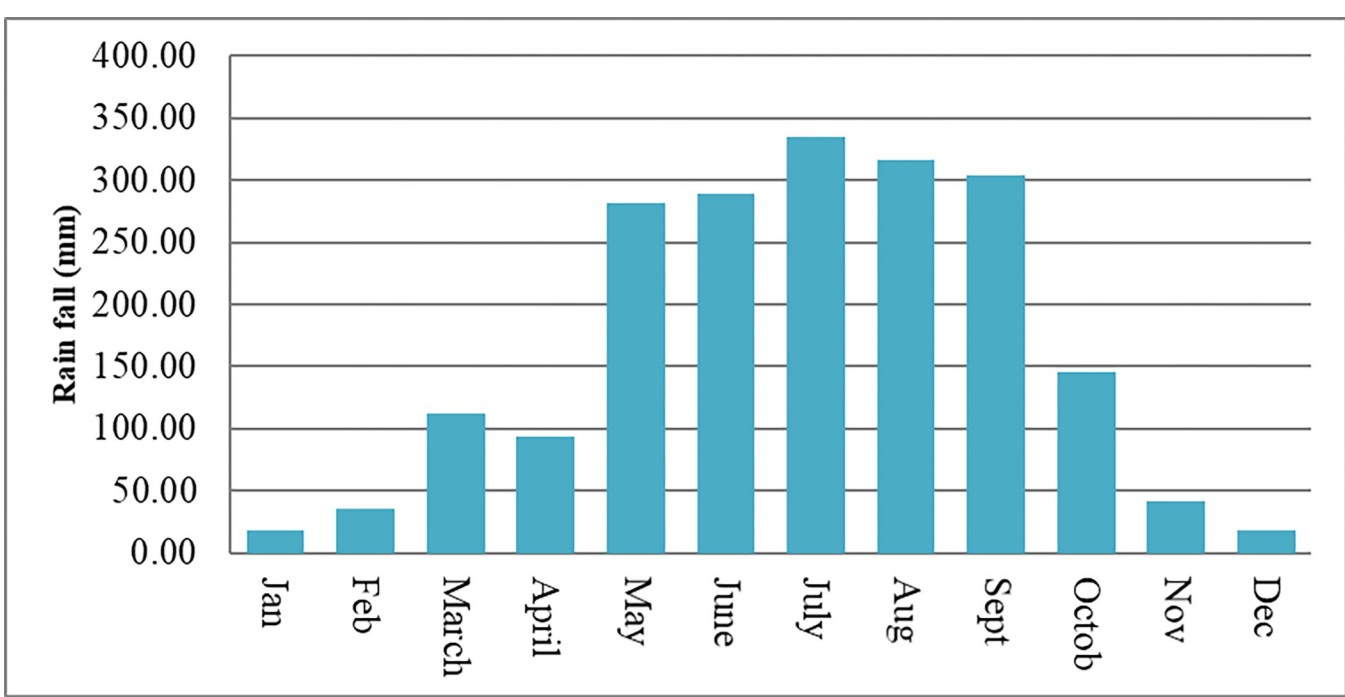

**Fig 2. Monthly total rainfall (mm) of Mettu during crop growth period in 2017.**

weight at 105˚C, of the soil samples was used to determine bulk density [17]. Five random disturbed soil samples (0-15cm depth) were collected diagonally and composite soil sample was made.

The composite sample was used for soil physiochemical analysis, and for the determination of lime requirement of the soil. The disturbed soil samples were air dried and sieved to pass through 2 mm sieve, and placed in a labeled plastic bag. Then, the samples were transported to Jimma Agricultural soil and plant tissue analysis laboratory. The soil sample were analyzed for particle size distribution(soil texture), which was done by Bouyoucos hydrometer method as [18]; while soil exchangeable acidity, exchangeable bases, soil pH, organic carbon(OC), total nitrogen(TN), available phosphorus and cation exchange capacity (CEC) for soil chemical analysis were selected. Available soil P and exchangeable acidity were determined using Bray-II method, as described by [19, 20] respectively.

After harvesting soil sample were taken from lime treated and untreated separately. The collected samples were air dried and sieved to pass through 2 mm sieve and submitted to soil laboratory for soil chemical properties analysis. Organic matter was determined using wet oxidation; Total N was determined by Kjeldahl method, as described by [21]. Cation exchange capacity (CEC) and exchangeable bases (Ca, Mg, K and Na) were determined ammonium acetate at pH 7. The potential cation exchange capacity (CEC) of the soil was determined from the NH4+ saturated samples that were subsequently replaced by K+ using KCl solution. The excess salt was removed by washing with ethanol and the NH4+ that was displaced by K+ was measured using the micro-Kjeldahl procedure [22], and reported as CEC. Exchangeable Ca and Mg was analyzed using Atomic Absorption spectrophotometer (AAS). Exchangeable Na and K were analyzed using flame photometer as described by [22].

Available soil P was determined using Bray-II method, as described by Bray and Kurtz (1945). The soil pH was determined in soil water suspension of 1:2.5 (soil: water ratio) using pH meter, as described by Van Reeuwijk (1992) [23]. Exchangeable acidity was determined by

**Table 1. Physicochemical properties of the experimental soil prior to sowing and after harvesting.**

| Parameters | Before sowing | after harvesting | |
|---|---|---|---|
| Particle size distribution | | Limed | Unlimed |
| Clay (%) | 49.00 | | |
| Sand (%) | 38.00 | | |
| Silt (%) | 13.00 | | |
| Textural class | Clayey | | |
| pH (H2O) | 4.400 | 4.73 | 4.48 |
| Exchangeable acidity (cmol(+)/kg) | 2.720 | 1.52 | 2.41 |
| Exchangeable Al (cmol(+)/kg) | 1.460 | 0.93 | 1.38 |
| Organic carbon (%) | 2.210 | 2.45 | 2.22 |
| CEC (cmol (+) kg$^{-1}$) | 18.75 | 21.04 | 18.89 |
| Total N (%) | 0.210 | 0.24 | 0.22 |
| Available P(BrayII) (mg kg$^{-1}$) | 2.950 | 4.39 | 2.98 |
| Exchangeable K (cmol (+) kg$^{-1}$) | 0.330 | 0.67 | 0.40 |
| Exchangeable Ca (cmol(+) kg$^{-1}$) | 3.550 | 5.39 | 3.81 |
| Exchangeable Mg (cmol(+) kg$^{-1}$) | 1.380 | 1.59 | 1.40 |

saturating the soil samples with potassium chloride solution and titrates with sodium hydroxide as described by [13]. From the same extractant, exchangeable Al in the soil was titrated with a standard solution of 0.02M HCl.

## Determination of lime requirements

The amount of lime applied was determined based on the exchangeable acidity, mass per 0.15m furrow slice and bulk density of the soil) [24], considering the amount of lime needed to neutralize the acid content (Al + H) of the soil up to the permissible acid saturation level for soybean growth.

$$LR, \frac{CaCo3kg}{ha} = \frac{Cmol \frac{EA}{Kg} \text{ of } soil * 0.15m * 10m^2 * BD\left(\frac{g}{cm^3}\right) * 1000 * crop\ factor}{2000}$$

Where: BD = bulk density, EA = exchangeable acidity (exch. H$^+$ + Al$^{3+}$), LR = lime requirement, 0.15m = plough depth/depth of lime incorporation.

## Planting material, treatments, experimental design and procedures

The treatments comprised of two factors namely; two soil amendments (control or no lime and limed) and fifteen different soybean genotypes (Table 2). The treatments were laid out in split plot design. The experiment has three technical replications and six biological replication in 4m by 2.4 m (9.6 m$^2$) plot size. Per row eighty (80) and per plot/treatments 320 seeds were used. Soil amendments (lime and unlimed) applied as main plot treatments and soybean genotypes were applied to sub plot treatments. The different soybean genotypes for the trial were identified from previous advanced Multi-Location Yield Trials, including previous soil acidity tolerance screening trials.

The lime requirement (LR) of the soil for the plots was determined based on exchangeable acidity (EA) or acid saturation of the experimental soil. The lime rate was, therefore, 3457.8 kg/ha based on exchangeable acidity of the soil. Calcium carbonate (CaCO3) was used as the source of lime and the whole doses of lime were broadcasted on limed plots manually, uniformly and mixed in the top 15 cm soil layer, a month before sowing. Reduction percentage

**Table 2. Soybean genotypes used for the experiment.**

| Genotypes | Back ground information and Source |
|---|---|
| JM-CLK/CRFD-15-SA | Inbreed line identified from local crosses at JARC |
| JM-ALM/PR142-15-SC | Inbreed line identified from local crosses at JARC |
| JM-ALM/H3-15-SC-1 | Inbreed line identified from local crosses at JARC |
| BRS 268 | Introduced from Brazil |
| JM-HAR/DAV-15-SA | Inbreed line identified from local crosses at JARC |
| JM-CLK/G99-15-SC | Inbreed line identified from local crosses at JARC |
| JM-CLK/G99-15-SB | Inbreed line identified from local crosses at JARC |
| JM-H3/SCS-15-SG | Inbreed line identified from local crosses at JARC |
| Pl 567046A | Introduced from USA |
| SCS-1 | Pipe line from Pawe agricultural research Center |
| Pl 423958 | Introduced from USA |
| H-7 | Pipe line from Mozambique ARI |
| HAWASSA-04 | Released variety from Hawassa Agricultural research |

JARC = Jimma Agricultural Research Center

for grain yield, root volume, growth and nodulation parameter was calculated as the ratio in lime treated to lime untreated soil, which also showed higher differences among the tested genotypes. The seeds were sown in rows to maintain plant to plant distance of 5 cm and 60 cm between rows. The cropping was done in main season by rain fall without any germination and seedling establishment problem. Cultural practices (i.e. weeding, hoeing etc, the experimental field was weeded by hand five times during the growing period uniformly for all treatments).

## Statistical analysis

The data was subjected to analysis of variance (ANOVA) using Statistical Analysis System (SAS) software version 9.3 [25] using proc GLM procedure. The difference between treatment means was separated using LSD 5% value. Correlation analysis between the traits was carried out to determine the magnitude and degree of their associations.

## Results and discussion

### Effects of soil acidity on yield and yield components of soybean genotypes

Genotypes reflected significant differences for number pods per plant, number of seeds per plant, grain yield, and hundred seed weight and above ground biomass in both limed and unlimed soil regimes (Table 3). The interaction of amendment*genotypes was also highly significant (p≤0.01) for all yield components and yield, however hundred seed weight was only affected by the main effect of genotypes and amendments.

The response of the observed soybean characters in acidic soil varied among genotypes. Grain yield, number of pods per plant, number of seeds per plant, hundred seed weight and above ground biomass in unlimed soil with low pH was lower than in treated soil or same with that in limed (Table 4). Even, grain yield of BRS268 genotype under unlimed acid soil were higher than in limed acid soil.

On average, the genotypes gave higher yield and yield components in lime treated soil (Table 4). PI567046A genotype gave higher grain yield, above ground biomass, number of pods and seeds per plant in lime treated acid soil; while genotype SCS-1 gave the lowest yield and genotype PI423958 gave the lowest number of pods, seeds and above ground biomass in

**Table 3. Mean square of growth, root, nodulation, yield and yield components of soybean genotypes grown on limed and unlimed soil on field.**

| Source of variations | Gen | Lime | Lime*Gen | Error (a) | Total |
|---|---|---|---|---|---|
| Parameters | | | | | |
| Yield | 582515.8** | 341225.8** | 98147.32** | 4211.35 | 10148754.36 |
| No of pod per plant | 195.62** | 239.44** | 39.59** | 0.41346 | 3558.5 |
| No of seed per plant | 855.29** | 801.62** | 144.54** | 3.444 | 15005.62 |
| Above ground biomass | 5.275** | 5.436** | 1.3597** | 0.1852 | 109.9 |
| Number of nodule | 206.31** | 1626.47** | 55.06** | 1.143 | 5350.77 |
| Root dry weight | 0.093** | 0.2722** | 0.01715** | 0.001716 | 1.9161 |
| Plant height | 810.97** | 577.85**** | 7.4215** | 2.143 | 12190.7 |
| Root volume | 1.0074** | 4.71512** | 0.2408ns | 0.227 | 35.624 |
| Shoot dry weight | 7.3028** | 40.1735** | 2.141** | 0.0228 | 173.756 |
| Hundred seed weight | 19.486** | 10.64** | 0.739ns | 3.122 | 492.733 |

** = highly significant different at 1% level of significance, ns = non-significant d/t at 5% level of significance, Gen = genotypes

lime untreated acid soil (Table 4). Under lime untreated soil condition, the maximum grain yield and above ground biomass was obtained from variety HAWASSA-04, and genotypes BRS268 and PI567046A gave highest number of pods and seeds per plant respectively. Under unlimed acid soil condition, the highest increasing percentage for yield and above ground biomass, was shown by BRS268 and JM-DAV/PR142-15-SA respectively, while the highest decreasing percentage was shown by PI567046A for both yield and above ground biomass.

The yield increments with lime application might be due to the probability of obtaining the available P from decomposed OM by microorganisms, when the pH value of the soil improved

**Table 4. Interaction effect of genotypes and lime for yield and yield components of soybean genotypes grown under limed and unlimed soil at Mettu during 2017 main cropping season.**

| Genotypes | Yield (kg/ha) | | NPPP | | NSPP | | AGB (ton/ha) | |
|---|---|---|---|---|---|---|---|---|
| | L | UL | L | UL | L | UL | L | UL |
| PI567046A | 1943.93[a] | 1069.87[d-i] | 47.1[a] | 26.6[cd] | 92.33[a] | 55.87[c] | 7.02[a] | 3.23[e-k] |
| HAWASSA-04 | 1576.77[ab] | 1553.11[bc] | 29.36[bc] | 22.93[gh] | 60.73[b] | 41.53[e-i] | 5.06[b] | 4.2[bc] |
| JM-PR142/H3-15-SB | 1328.29[b-d] | 1027.24[e-j] | 21.53[h-k] | 19.46[l-n] | 43.67[ef] | 37.13[j-l] | 4.24[bc] | 3.58[c-g] |
| BRS268 | 1143.47[d-g] | 1319.83[b-e] | 30.33[b] | 27.6[cd] | 47.73[d] | 47.73[d] | 4.01[cd] | 3.79[c-f] |
| JMALM/PR142-15-SC | 1214.46[c-f] | 1121.35[d-g] | 20.27[k-m] | 21.33[i-k] | 44.9[de] | 44.67[de] | 4.01[c-e] | 3.96[c-e] |
| JM-H3/SCS-15-SG | 956.49[f-k] | 1096.45[d-h] | 20.53[j-l] | 20.46[j-m] | 44.36[de] | 43.53[ef] | 3.52[c-h] | 3.57[c-g] |
| JM-CLK/G99-15-SB | 1076.24[d-h] | 756.98[j-p] | 22.93[gh] | 21.8[h-j] | 38.9[i-k] | 39.13[h-j] | 2.79[h-m] | 2.65[j-n] |
| JM-CLK/CRFD-15-SA | 935.05[f-l] | 643.49[m-p] | 22.53[hi] | 18.07[m-o] | 40.33[f-j] | 33.47[l-n] | 3.43[d-j] | 2.89[g-m] |
| JM-DAV/PR142-15-SA | 934.71[f-l] | 915.36[g-m] | 24.33[fg] | 24.8[ef] | 42.76[e-h] | 42.26[e-i] | 2.99[g-m] | 3.07[f-l] |
| H-7 | 772.48[j-p] | 821.79[h-n] | 21.33[i-k] | 18nop | 39.67[g-j] | 36.87[j-l] | 2.44[l-n] | 2.27[m-o] |
| JM-CLK/G99-15-SC | 783.83[i-o] | 818.21[h-n] | 22.53[hi] | 21.93[h-j] | 43.13[e-g] | 43.07[e-g] | 3.51[c-i] | 3.52[c-h] |
| SCS-1 | 618.95[n-p] | 510.49[p] | 17.93[n-p] | 16.53[p] | 32.16[mn] | 29.8[n] | 2.58[k-n] | 2.36[l-n] |
| JM-HAR/DAV-15-SA | 737.46[k-p] | 690.96[k-p] | 19.27[l-n] | 17.4[op] | 34.33[lm] | 31.87[mn] | 2.35[l-n] | 2.23[m-o] |
| JM-ALM/H3-15-SC-1 | 653.17[m-p] | 637.54[n-p] | 21.27[i-k] | 18.93[m-o] | 39.1[ijk] | 35.27[k-m] | 3.06[f-l] | 2.73[h-m] |
| PI423958 | 682.82[l-p] | 528.21[o] | 13.33[q] | 9.76[r] | 22.13[o] | 14.33[p] | 1.87[no] | 1.43[o] |
| Mean | 1023.87 | 900.73 | 23.63 | 20.37 | 44.40 | 38.43 | 3.52 | 3.04 |
| CV | 6.74 | | 2.922 | | 4.48 | | 13.12 | |

Means with the same letters are not significantly different at 5% level of significance. UL- unlimed; L- Limed; NPPP- Number of pod per plant; NSPP–Number of seed per plant; AGB- above ground biomass; CV- coefficient of variation

due to liming, which might have resulted in increased grain yield. Liming also improved the ability of the plant to absorb P, when Al toxicity has been eliminated, and enhanced the vegetative growth of soybean genotypes, which resulted in increased dry biomass yield. In line with this result, [26] also reported that the highest barley grain yield was obtained under the application of 2.2 t/ha lime than unlimed acid soil. The genotypes responded to the applied lime for number of pod and seed, which might be due to lime enhanced vegetative growth and make genotypes to bear higher number of pod than lime untreated acid soil and also lime is neutralized acid soil which might increase the availability of phosphorus for plant uptake by reducing phosphorus fixation on acid soil. Lime also improved soil pH and enhanced growth and yield of soybean genotypes, as a result of increased P availability, photosynthesis intensity, flowering, seed formation and fruiting of the crops also increased (Kisinyo et al., 2016) [33]. In line with this results [13] reported that lime application increased a number of pod and seed per plant. [27] also reported that the application of lime produced the highest seeds per plant than unlimed soil. [28] reported 36.4% plant height decrease in soybean on unlimed acid soil compared with limed acid soil which has a direct relationship with yield increment under limed soil conditions.

The highest hundred seed weight was recorded for PI423958 genotype under lime treated soils and the lowest hundred seed weight was recorded for H-7 under lime untreated soil (Tables 5 and 6). In this study, the variable of tested genotypes has been observed, which indicates the presence of difference among the tested genotypes for hundred seed weight. Application of lime didn't affect hundred seed weight of genotypes (Table 5). In agreement with this result, [29] reported significant difference among soybean genotypes for hundred seed weight, in which the highest hundred seed weight was produced by BFS 39 genotype and the lowest hundred seed weight was recorded from Roba. [30] reported non- significant effect of liming on hundred seed weight of common bean. In general; lime application to the soil increased yield, number of pod and seed and above ground biomass of soybean genotypes by about 13.67, 16.06, 15.53 and 16.18% respectively, however hundred seed weight were not affected by lime.

Hundred seed weight is higher in un-limed soil, this might due to the improvement of soil pH in response to lime amendment, which enhanced growth and yield of the plant, as a result of increased availability of P that might have increased intensity of photosynthesis, flowering, seed formation and fruiting, as a result these formed fruit is competed for nutrient to fulfill the seed and the seed size become decreased which have the direct effect on seed weight.

### Effect of soil acidity on root, growth, nodulation parameter of soybean genotypes

There were highly significant (P<0.01) differences among genotypes for root dry weight, number of nodule, plant height and shoot dry weight in both soils regimes. The interaction of lim

**Table 5. Average values of grain yield (kg/ha), NPPP, NSPP, AGB (ton/ha) and HSW (g) of soybean genotypes grown under limed and unlimed acid soil at Mettu during 2017.**

| Treatments | Yield (kg/ha) | NPPP | NSPP | AGB(ton/ha) | HSW(g) |
|---|---|---|---|---|---|
| Limed | 1023.87[a] | 23.63[a] | 44.40[a] | 3.525[a] | 13.34[a] |
| Unlimed | 900.73[b] | 20.36[b] | 38.43[b] | 3.033[b] | 14.34[a] |
| PR | **13.67** | **16.06** | **15.53** | **16.22** | **-6.97** |

Means with the same letters are not significantly different at 5% level of significance. NPPP- Number of pod per plant; HSW = hundred seed weight, NSPP–Number of seed per plant; AGB- above ground biomass, PR = percent of reduction

**Table 6. Main effect of soybean genotypes for hundred seed weight grown under acid soil at Mettu on field during 2017 main cropping season.**

| Sub-plot treatments (Genotype) | HSW (g) |
|---|---|
| PI423958 | 17.5[a] |
| JMALM/PR142-15-SC | 16.43[ab] |
| HAWASSA-04 | 15.08[bc] |
| JM-H3/SCS-15-SG | 15.078[bc] |
| JM-PR142/H3-15-SB | 14.68[bc] |
| BRS268 | 14.46[bc] |
| JM-CLK/CRFD-15-SA | 14.228[c] |
| JM-HAR/DAV-15-SA | 13.83[c] |
| JM-CLK/G99-15-SB | 13.66[c] |
| JM-CLK/G99-15-SC | 13.39[c] |
| JM-ALM/H3-15-SC-1 | 13.35[c] |
| SCS-1 | 13.31[c] |
| JM-DAV/PR142-15-SA | 13.18[c] |
| PI567046A | 11.08[d] |
| H-7 | 10.53[d] |
| CV | 12.617 |

Means with the same letters are not significantly different at 5% level of significance. HSW = hundred seed weight, CV = Coefficient of variation

ing*genotypes was also highly significant($p \leq 0.01$) for root dry weight, number of nodule, plant height and shoot dry weight, except for root volume only the main effect of liming and genotype s were significant. Growth, root and nodulation parameters of soybean genotypes grown under lime treated and untreated soils are indicated in (Tables 7 and 8). On average, the genotypes gave higher root, growth and nodulation parameters under lime treated acid soil (Tables 7 & 8). These results signified that application of lime increasing root, growth and nodulation parameters. JMALM/PR142-15-SC and BRS268 genotype gave higher root volume and root dry weight in both limes treated and untreated soil, and indicating these genotypes might be among acidic soil tolerant genotypes; while genotype PI423958 gave the lowest root dry weigh on lime untreated acid soil (Table 8). [31] reported that among the fifteen soybean genotypes tested MLGG 0064 genotype showed the highest root dry weight under the control soil condition (pH 7), while the lowest root length was shown by genotype MLGG 0377 in Mn toxicity condition, which shows varietal difference for acid soil adaptation.

The alteration of root dry weight and root volume includes decreasing and increasing of the root length and root hair. Decrease percentage of root dry weight and root volume in acid soil stress conditions varied among the tested genotypes. Limed soil condition showed the highest root dry weight and root volume than in unlimed soil. This might be due to liming improved the P uptake capacity of plants which facilitate root growth, and then increased root diameter or root thickness of the genotypes, and root dry weight is the result of root growth and development, including root length and number of lateral roots. Alteration in root length and number of roots causes an alteration in root dry weight and root volume. Alteration also occurs on root hair length and root hair density [32] that cause an alteration in root dry weight.

There was no negative values existed for root dry weight in unlimed acid soil conditions (Table 10), but negative value existed for number of nodule. Negative value suggests an increasing variable from the optimal or relatively optimal condition to the severer condition. The highest number of nodule was obtained from JM-DAV/PR142-15-SA; while the lowest

**Table 7. Interaction effect of genotypes and lime for SDW, NN and PHT of soybean genotypes grown under limed and unlimed soil at Mettu during 2017 main cropping season.**

| Genotypes | SDW (g/plant) | | NN/plant | | PHT (cm) | |
|---|---|---|---|---|---|---|
| | L | UL | L | UL | L | UL |
| PI567046A | 6.32$^{cd}$ | 6.32$^{cd}$ | 32.56$^{f}$ | 20$^{m}$ | 83.73$^{a}$ | 73.267$^{b}$ |
| HAWASSA-04 | 7.045$^{ab}$ | 7.045$^{ab}$ | 39.067$^{b}$ | 33.6$^{e}$ | 55.40$^{c}$ | 50.33$^{cdef}$ |
| JM-PR142/H3-15-SB | 5.40$^{hij}$ | 3.99$^{no}$ | 39.40$^{b}$ | 32.27$^{fg}$ | 54.72$^{cd}$ | 47.00$^{efghi}$ |
| BRS268 | 5.93$^{defg}$ | 5.95$^{def}$ | 31.23$^{ghi}$ | 31.26$^{ghi}$ | 53.80$^{cde}$ | 48$^{defgh}$ |
| JMALM/PR142-15-SC | 6.16$^{cde}$ | 5.45$^{ghij}$ | 37.33$^{c}$ | 30.87$^{hi}$ | 46.27$^{fghi}$ | 44.2$^{fghijk}$ |
| JM-H3/SCS-15-SG | 5.58$^{fghi}$ | 4.46$^{lmn}$ | 35.20$^{d}$ | 26.2$^{j}$ | 48.93$^{cdefg}$ | 44.87$^{fghij}$ |
| JM-CLK/G99-15-SB | 7.56$^{a}$ | 4.46$^{lmn}$ | 37.33$^{c}$ | 23.33$^{kl}$ | 43.067$^{ghijk}$ | 40.67$^{jklm}$ |
| JM-CLK/CRFD-15-SA | 4.34$^{mn}$ | 2.92$^{pq}$ | 37.13$^{c}$ | 22.67$^{l}$ | 48.93$^{cdefg}$ | 42.73$^{ghijk}$ |
| JM-DAV/PR142-15-SA | 4.76$^{klm}$ | 4.78$^{klm}$ | 32.40$^{f}$ | 31.67$^{fg}$ | 37.33$^{klmno}$ | 34.40$^{lmno}$ |
| H-7 | 5.18$^{ijk}$ | 4.027$^{no}$ | 39.00$^{b}$ | 34.13$^{de}$ | 32.00$^{nop}$ | 29.60$^{op}$ |
| JM-CLK/G99-15-SC | 5.73$^{efgh}$ | 4.80$^{klm}$ | 30.33$^{i}$ | 23.2$^{kl}$ | 48.8$^{cdefg}$ | 43.13$^{ghijk}$ |
| SCS-1 | 3.60$^{o}$ | 1.97$^{r}$ | 24.20$^{k}$ | 15.67$^{o}$ | 44.83$^{fghij}$ | 39.0$^{jklmn}$ |
| JM-HAR/DAV-15-SA | 4.83$^{kl}$ | 4.97$^{jk}$ | 31.23$^{ghi}$ | 30.8$^{hi}$ | 38.6$^{jklmn}$ | 33.60$^{mnop}$ |
| JM-ALM/H3-15-SC-1 | 2.94$^{p}$ | 2.38$^{qr}$ | 34.36$^{de}$ | 17.6$^{n}$ | 45.87$^{fghi}$ | 41.50$^{hijkl}$ |
| PI423958 | 6.48$^{bc}$ | 3.69$^{o}$ | 55.27$^{a}$ | 35.16$^{d}$ | 32.39$^{nop}$ | 26.20$^{p}$ |
| | 5.45 | 4.12 | 35.72 | 27.22 | 47.63 | 42.56 |
| CV | 3.15 | | 3.39 | | 3.245 | |

Means with the same letters are not significantly different at 5% level of significance. UL- unlimed; L- Limed; SDW- shoot dry weight; NN–number of nodule; PHT-plant height; CV- coefficient of variation

**Table 8. Main effect of soybean genotypes for RV and interaction effect of lime and genotypes for RDW grown under limed and unlimed acid soils at Mettu on field during 2017 main cropping season.**

| Genotypes | RDW (g/plant) | | V/plant in ml |
|---|---|---|---|
| | Limed | Unlimed | |
| PI567046A | 0.75$^{cde}$ | 0.433$^{j}$ | 2$^{bcde}$ |
| HAWASSA-04 | 0.81a-d | 0.807$^{abcd}$ | 2.33$^{bc}$ |
| JM-PR142/H3-15-SB | 0.79$^{bcd}$ | 0.74$^{cde}$ | 2.5$^{b}$ |
| BRS268 | 0.84a-d | 0.83$^{abcd}$ | 2.33$^{bc}$ |
| JMALM/PR142-15-SC | 0.937$^{a}$ | 0.873$^{abc}$ | 3.17$^{a}$ |
| JM-H3/SCS-15-SG | 0.71d-g | 0.637$^{efgh}$ | 2.167$^{bc}$ |
| JM-CLK/G99-15-SB | 0.647$^{efgh}$ | 0.44$^{j}$ | 1.5$^{e}$ |
| JM-CLK/CRFD-15-SA | 0.893$^{ab}$ | 0.75$^{cde}$ | 2.43$^{b}$ |
| JM-DAV/PR142-15-SA | 0.73$^{def}$ | 0.72$^{def}$ | 1.87$^{cde}$ |
| H-7 | 0.637$^{efgh}$ | 0.58$^{ghi}$ | 1.833$^{cde}$ |
| JM-CLK/G99-15-SC | 0.657$^{efgh}$ | 0.47$^{ij}$ | 1.6$^{de}$ |
| SCS-1 | 0.8$^{abcd}$ | 0.47$^{ij}$ | 2.1$^{bcd}$ |
| JM-HAR/DAV-15-SA | 0.55$^{hij}$ | 0.55$^{hij}$ | 2$^{bcde}$ |
| JM-ALM/H3-15-SC-1 | 0.717$^{defg}$ | 0.657$^{efgh}$ | 2.033$^{b-e}$ |
| PI423958 | 0.59$^{fghi}$ | 0.427$^{j}$ | 1.833$^{cde}$ |
| Mean | **0.735** | **0.625** | **CV = 22.55** |
| CV | **6.093** | | |

Means with the same letters are not significantly different at 5% level of significance. RDW- root dry weight; RV–root volume; CV- coefficient of variation

**Table 9. Average values of SDW (g/plant), NN/plant, PHT, RDW (g/plant) and RV (g/plant) of soybean genotypes grown under limed and unlimed acid soil at Mettu.**

| Treatments | SDW (g/plant) | NN/plant | PHT (cm) | RDW (g/plant) | RV /plant |
|---|---|---|---|---|---|
| Limed | 5.46[a] | 35.72[a] | 47.64[a] | 0.735[a] | 2.34[a] |
| Unlimed | 4.12[b] | 27.22[b] | 42.57[b] | 0.625[b] | 1.88[b] |
| PR | **32.52** | **31.23** | **11.91** | **17.6** | **24.46** |

Means with the same letters are not significantly different at 5% level of significance. RDW- root dry weight; RV- root volume, SDW-shoot dry weight, PHT-plant height, NN-number nodule; PR = percent of reduction; CV- coefficient of variation

number of nodule was recorded from SCS1 genotype. This might be due to liming effect on nodule weight and nodule numbers. There was one genotype showing an increase in number of nodules in unlimed ed acid soil condition. In line with this finding [11] reported two soybean genotypes i.e., H3 and PR-142 [15] showed the highest number of nodules per plant at 100 kg ha$^{-1}$P with lime and Essex-1 genotype showed the lowest number of nodules per plant at lime untreated plot among the other tested genotypes. The highest plant height was recorded for PI567046A genotype both under lime treated and untreated acid soils. On the other hand, the shortest plant height was recorded for PI423958genotype under lime untreated soil. This indicated that genotypes responded to liming, which might be due to the effect of liming that neutralized soil acidity, which in turn might have improved the availability of plant nutrients, particularly phosphorus and calcium and lowered the concentration of toxic cations, mainly Al$^{3+}$ ions. The results are similar with the results of [33] who reported that a growth of plant is increased on acid soil in response to the application of lime.

The highest shoot dry weight was obtained from JM-CLK/G99-15-SB genotype, while the lowest shoot dry weight were recorded from SCS-1 genotype (Table 7). The reduction of shoot dry weight under the control or unlimed acidic soil condition might be due to Al toxicity, and

**Table 10. Decrease percentage of yield, above ground biomass, number of pod and seed per plant, number of nodules, root dry weight and plant height of some soybean genotypes under unlimed acid soil conditions compared with limed acid soil.**

| Genotypes | YLD | AGB | NN | PHT | RDW | NPPP | NSPP |
|---|---|---|---|---|---|---|---|
| HAWASSA-04 | 1.50 | 17.0 | 13.9 | 9.13 | 0.00 | 21.6 | 31.6 |
| PI567046A | 44.96 | 54.0 | 38.4 | 12.51 | 42.7 | 43.4 | 39.5 |
| PI423958 | 22.64 | 23.4 | 36.5 | 18.8 | 27.1 | 27.4 | 35.26 |
| JMALM/PR142-15-SC | 7.67 | 1.25 | 17.3 | 4.39 | 7.40 | -5.2 | 0.52 |
| JM-HAR/DAV-15-SA | 6.31 | 5.11 | 1.39 | 12.95 | 0.00 | 9.69 | 7.18 |
| JM-PR142/H3-15-SB | 22.67 | 15.5 | 18.1 | 14.1 | 6.30 | 9.63 | 14.96 |
| H-7 | -6.38 | 6.84 | 12.4 | 7.5 | 7.90 | 15.6 | 7.06 |
| BRS268 | -15.4 | 5.41 | -0.1 | 10.78 | 1.20 | 9.03 | 0.00 |
| JM-H3/SCS-15-SG | -14.6 | -1.3 | 25.5 | 8.32 | 11.3 | 0.36 | 1.66 |
| JM-CLK/CRFD-15-SA | 31.18 | 15.7 | 38.9 | 12.67 | 15.7 | 19.8 | 17.02 |
| JM-ALM/H3-15-SC-1 | 2.40 | 10.7 | 48.6 | 9.43 | 8.50 | 10.9 | 9.72 |
| JM-CLK/G99-15-SC | -4.39 | -0.3 | 23.5 | 11.61 | 27.7 | 2.66 | 0.15 |
| SCS-1 | 17.53 | 8.41 | 35.2 | 13.02 | 46.3 | 7.81 | 7.07 |
| JM-CLK/G99-15-SB | 29.66 | 4.91 | 37.5 | 5.52 | 31.3 | 4.97 | -0.51 |
| JM-DAV/PR142-15-SA | 2.06 | -2.5 | 2.26 | 7.87 | 2.70 | -1.9 | 1.17 |

Where, NPPP = number pod per plant, NSPP = number of seeds per plant, PHT = plant height, RDW = root dry weight, YLD = yield, NN = number of nodules per plant, AGB = above ground biomass

low Ca, Mg and P concentrations in the shoot, which resulted in decreased photosynthetic capacity that directly affected, shoot growth and developments. This alteration was also due to the low pH inhibits root growth; reduce $Ca^{2+}$ and $Mg^{2+}$ in the leaf and reduce rhizobia activity to f orm nodule [34] as well as the Mn toxicity. Different result was reported; [35] reported that decreasing solution pH and Ca concentration decreased the shoot dry weight. However, plant height, root volume, root dry weight, number of nodule and shoot dry weight is also affected by the genotypes. Lime applications to acid soil increased plant height, root volume, root dry weight, number of nodule and shoot dry weight of soybean genotypes by about 11.91, 24.47, 17.6, 31.22 and 32.5% respectively (Tables 9 and 10).

## Correlation analysis

Grain yield was significantly (P$\leq$ 0.01) and positively correlated with all root parameters viz., root dry weight and root volume and with all growth parameters viz., plant height and shoot dry weight and also with number of nodule at both limed and unlimed soil (Table 11). The significant and positive correlations of grain yield with the rooting parameters viz., root volume and root dry, under acid soil condition or under unlimed acid soil (hydrogen and aluminum toxicity) indicates the importance of the root parameters for acid soil tolerance. This also implies that selection for acid soil tolerance should consider these important root parameters. Similar to this finding [11] also reported the significant and positive associations of soybean grain yield with its root characters like root volume, root dry and fresh weight.

Grain yield is the product of its yield components, such as number of pods per plant, number of seeds per plant and above ground dry biomass were highly significant and positively correlated with its grain yield at both lime treated and untreated acid soil (Table 11). However, grain yield was strongly correlated with above ground biomass (r = 0.90), followed by number of seeds (r = 0.87) and number of pods per plant (r = 0.82) at limed soil among yield parameters, respectively. Other authors, such as [36, 37] reported that the significant associations of barley grain yield with its yield components. Results obtained in this study on soil treated with lime clearly showed that the remarkable increase in number of pods and seeds per plant, and greatly contributed to increase in grain yield of soybean. The negative correlation of number of nodules with number of seed and pod under unlimed soil (Table 11) indicates the competitiveness of these traits.

## Conclusion

For the conclusion, the observed characters showed a different response in acid soil toxicity. The fifteen genotypes responded differently to acid soil. A preliminary field screening of soybean genotypes in south western Ethiopia has demonstrated the presence of genetic variability among genotypes in tolerating soil acidity stress. The observed characters of the sensitive genotypes decreased, while the tolerant genotypes could remain stable or increased. Root dry weight, root volume, number of nodule, plant height, shoot dry weight, grain yield, biomass, number of pod per plant, number of seed per plant, and hundred seed weight under unlimed soil were lower than in limed soil condition. However, not all these characters always decrease in unlimed soil condition. Their increments of grain yield in unlimed soil were found for BRS268 genotype. Genotype of BRS268 was the tolerant genotypes based on the reduction percentage of selected parameters. These results also give a clear indication that the grain yield was very closely associated with number of pods per plant, seed per plant, root dry weight, root volume, and shoot dry weight and number of nodule in both unlimed and limed soil. It seems that these parameters are useful characters to select for high yield in soybean breeding programs for soil acidity tolerance. Studying responses of selected genotypes with contrasting

**Table 11. Pearson correlation analysis for growth, root, nodulation, yield and yield components of soybean genotypes grown under lime treated (1st) and lime untreated (2nd) soil on field at Mettu.**

| | YLD | PHT | NSPP | NPPP | AGB | SDW | RV | RDW | NN |
|---|---|---|---|---|---|---|---|---|---|
| YLD | 1 | | | | | | | | |
| PHT | 0.82** | 1 | | | | | | | |
| NSPP | 0.87** | 0.90** | 1 | | | | | | |
| NPPP | 0.82** | 0.87** | 0.95** | 1 | | | | | |
| AGB | 0.90** | 0.91** | 0.92** | 0.86** | 1 | | | | |
| SDW | 0.56** | 0.21ns | 0.31* | 0.28$^{ns}$ | 0.31* | 1 | | | |
| RV | 0.15$^{ns}$ | 0.15ns | 0.04$^{ns}$ | -0.07$^{ns}$ | 0.22$^{ns}$ | -0.08$^{ns}$ | 1 | | |
| RDW | 0.37* | 0.38* | 0.27$^{ns}$ | 0.23$^{ns}$ | 0.43** | -0.07$^{ns}$ | 0.55** | 1 | |
| NN | 0.01$^{ns}$ | -0.29$^{ns}$ | -0.25$^{ns}$ | -0.29$^{ns}$ | -0.19$^{ns}$ | 0.43** | 0.07$^{ns}$ | -0.18$^{ns}$ | 1 |
| | YLD | PHT | NSPP | NPPP | AGB | SDW | RV | RDW | NN |
| YLD | 1 | | | | | | | | |
| PHT | 0.52** | 1 | | | | | | | |
| NSPP | 0.66** | 0.75** | 1 | | | | | | |
| NPPP | 0.68** | 0.67** | 0.92** | 1 | | | | | |
| AGB | 0.76** | 0.55** | 0.70** | 0.67** | 1 | | | | |
| SDW | 0.63** | -0.13ns | 0.24ns | 0.35* | 0.45** | 1 | | | |
| RV | 0.48** | 0.19ns | 0.21ns | 0.23ns | 0.51** | 0.34* | 1 | | |
| RDW | 0.59** | 0.09ns | 0.29ns | 0.36* | 0.61** | 0.51** | 1.0** | 1 | |
| NN | 0.39** | -0.36* | -0.15ns | -0.08ns | 0.07ns | 0.70** | 0.27ns | -0.18ns | 1 |

Where, NPPP = number pod per plant, NSPP = number of seeds per plant, PHT = plant, Height, SDW = shoot dry weight, RDW = root dry weight, YLD = yield, NN = number of nodules per plant, AGB = above ground biomass

tolerance to soil acidity may help in generating information that could be utilized by breeding programs aimed at developing aluminum-tolerant cultivars for areas where soil acidity remains a key environmental constraint to crop production. In conclusion identification and use of soybean genotypes that are tolerant to acid soil conditions of Southwestern Ethiopia is a very useful approach to ensure economic stability to many subsistence farmers, who cannot afford the application of liming materials practices.

## Supporting information

**S1 Table. Supporting information" files" raw data for each parameter.**
(DOC)

## Acknowledgments

The authors are grateful to Jimma agricultural research center for facilitating the research work and also acknowledge the staff members of Mettu research sub center for their technical assistance during the time of conducting the field experiment and data collection.

## Author Contributions

**Conceptualization:** Tolossa Ameyu Bedassa, Abush Tesfaye Abebe, Alemayehu Regassa Tolessa.

**Formal analysis:** Tolossa Ameyu Bedassa.

**Funding acquisition:** Abush Tesfaye Abebe, Alemayehu Regassa Tolessa.

**Investigation:** Tolossa Ameyu Bedassa, Abush Tesfaye Abebe, Alemayehu Regassa Tolessa.

**Methodology:** Tolossa Ameyu Bedassa, Abush Tesfaye Abebe, Alemayehu Regassa Tolessa.

**Project administration:** Tolossa Ameyu Bedassa, Abush Tesfaye Abebe.

**Software:** Tolossa Ameyu Bedassa.

**Supervision:** Tolossa Ameyu Bedassa, Abush Tesfaye Abebe, Alemayehu Regassa Tolessa.

**Validation:** Tolossa Ameyu Bedassa, Abush Tesfaye Abebe.

**Visualization:** Tolossa Ameyu Bedassa, Abush Tesfaye Abebe, Alemayehu Regassa Tolessa.

**Writing – original draft:** Tolossa Ameyu Bedassa.

**Writing – review & editing:** Tolossa Ameyu Bedassa.

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
