## [Decision Letter · Decision Letter 0]

24 Mar 2021

PONE-D-20-38977

Tolerance to soil acidity of soybean (Glycine max L.) genotypes under field conditions at Southwestern Ethiopia

PLOS ONE

Dear Dr. Bedassa,

Thank you for submitting your manuscript to PLOS ONE. After careful consideration, we feel that it has merit but does not fully meet PLOS ONE’s publication criteria as it currently stands. Therefore, we invite you to submit a revised version of the manuscript that addresses the points raised during the review process.

Although the topic of the ms is interesting, yet both the reviewers have raised serious concerns in different sections of the draft. The manuscript can be accepted after a through revision.

We look forward to receiving your revised manuscript.

Kind regards,

Saddam Hussain

Academic Editor

PLOS ONE

Journal Requirements:

"The authors are grateful to the financial grant of the Ethiopian institute of agricultural

research through the Jimma agricultural research center."

"NO, The funders had no role in study design, data collection and analysis, decision to publish, or preparation of the manuscript."

3. We note you have included tables to which you do not refer in the text of your manuscript. Please ensure that you refer to Tables 5 and 9-11 in your text; if accepted, production will need this reference to link the reader to the Table.

4. We noticed you have some minor occurrence of overlapping text with the following previous publications, which needs to be addressed:

- http://uvadoc.uva.es/handle/10324/11141 (Introduction, paragraph 2, sentences 2-3) (Introduction, paragraph 4, sentences 3-4)

- https://biodiversitas.mipa.uns.ac.id/D/D1801/D180133.pdf (Conclusion, paragraph 1, sentence 9)

In your revision ensure you cite all your sources, and quote or rephrase any duplicated text outside the methods section. Further consideration is dependent on these concerns being addressed.

5. Please include your tables only as part of your main manuscript and remove the individual files. Please note that supplementary tables should be uploaded as separate "supporting information" files.

Reviewers' comments:

Reviewer's Responses to Questions

**Comments to the Author**

1. Is the manuscript technically sound, and do the data support the conclusions?

Reviewer #1: No

Reviewer #2: Yes

2. Has the statistical analysis been performed appropriately and rigorously? 

Reviewer #1: No

Reviewer #2: Yes

3. Have the authors made all data underlying the findings in their manuscript fully available?

Reviewer #1: No

Reviewer #2: Yes

4. Is the manuscript presented in an intelligible fashion and written in standard English?

Reviewer #1: No

Reviewer #2: Yes

5. Review Comments to the Author

Reviewer #1: Broad Comments

It would be very helpful if the authors would include continuous line numbers to facilitate the review process. Due to a lack of line numbers, the authors should please refer to the highlights in the pdf in order to see the reviewer comments for each specific section of the manuscript.

One issue is that it is unclear how many biological replicates were used in the experiments done. The authors need to have done at least three biological replicates and three technical replicates for each assay. From the descriptions of the experiments, it is not clear at all what was done. If they only did three technical replicates, then all they measured was the accuracy of the techniques/assays used, not the accuracy of the biological phenomenon. Three biological replicates ensures that the biological effect seen is consistent within individuals of the species tested. They need to clearly describe what is considered as a single biological replicate and what is considered to be a technical replicate. For each table and figure caption/legend, they need to clearly state the number of biological and technical replicates used. This is extremely critical as the authors are doing experiments over multiple soybean genotypes.

For some of the experiments, the authors use 5 random plants of each genotype. This is an extremely limited amount of sampling given that they are drawing conclusions that apply to the genotype. This is even more problematic as the authors do not seem to have biological or technical replicates for the parameters that they tested for each genotype. This needs to be rectified in order to give confidence that the conclusions they provide can be applied to all the individuals within a specific genotype.

In several places, the work done has been unfortunately obscured by very poor English language communication. It is highly recommended that the authors seek the help of English language editors in order to improve the quality of the manuscript. The language errors are sometimes extensive and significantly hamper the ability of the reviewer to understand what is being conveyed. The authors need to edit significant portions of the manuscript for clarity. This will allow the significance of the work done to be properly highlighted.

There are also areas where the authors need to provide citations for the statements they make. The authors also need to provide the raw data used in the analyses in supplementary files.

Reviewer #2: The study has investigated the tolerance to soil acidity of soybean (Glycine max L.) genotypes under field conditions at Southwestern Ethiopia. It is believed that this paper will be found interesting to the reader of the journal. The paper has written excellently and necessary data collected and analyzed critically and described well. However, some of the comments mentioned here need consideration before publication.

Comment: The abstract can serve as a stand-alone document, which succinctly described. So, the abstract should cover the salient findings more critically. Some sentences are too long in abstract, methodology and results and discussion. Describe it briefly or split it another sentence if possible.

Comment: On what basis, Genotype BRS268 showed higher yield (1319.83 kg ha-1 ) under lime untreated acid soil than lime treated acid soil(1143.47 kg ha -1 ) and showed less reduction percentage for number of nodule, root weight and number of seeds per plant? Either it resistant? Is it previously identified/claimed?

Comment: The introduction is informative, precise, and comprised of relevant content. The literary structure of the introduction is also good. Citation is missing at the end of paragraph 1, 3, 4, etc.

Comment: In introduction, Line 15-17 and 34-37 are looking repeated.

Comment: Material and methods section looks fine, the used methods are well presented. The experimental design was appropriate and the applied protocols were used correctly. The authors organized very nice research. The methods are simple and sufficiently described. However, gave proper reference that from the soybean genotypes identified previously.

Comment: In Table 3. What does mean Gen? Specify it/mention if used it as abbreviated.

Comment: According to results, the genotype BRS268 gave better yield in un-limed soil when compared to lime treated soil. So therefore it means this variety would not be recommended in lime treated or high pH soil? Justify it. Conclude it in discussion section and conclusion.

Comment: In results you have mentioned and discussed only genotype BRS268 gave better yield in un-limed soil. While there are some other genotypes i.e. H-7 and JM-H3/SCS-15-SG are too exist but not mentioned and discussed?

Comment: Why Average values of hundred seed weight is higher in un-limed soil. Justify it. Gave it reason in discussion section.

Comment: In results, CV is given, however, at some parameters it looks too small and in some it exceeds too large that’s mean standard error (S.E) small or too large, respectively. Why? If possible, either it is better to provide S.E instead of Giving CV?

Comment: Either it will be better to analyze the data statistically separately lime treated and un-limed soil, instead of combine?

Comment: In this paper you have just mentioned and analysed toxicity of H+ and Al3+, Is there no any else toxicity i.e., Fe, Cu, Zn, Mn, etc., as common problem found in acidic soils.

Comment: The discussion section is also well described. Discussion should be appropriate in which results are discussed critically in response to support as well contradiction, instead of describing other cited results. Justify your results with some latest citations.

Comment: The citations used are somewhat old in introduction and discussion. Correlate your problem (introduction) and results with some of current latest citations.

Comment: Conclusions is presented in an appropriate fashion however, looking too long.

Comment: Check references section appropriately according to the journal requirement.

6. PLOS authors have the option to publish the peer review history of their article (what does this mean?). If published, this will include your full peer review and any attached files.

Reviewer #1: No

Reviewer #2: **Yes: **Dr. M. Asaad Bashir 

Assistant Professor,

Department of Soil Sciences,

Faculty of Agriculture & Environment,

The Islamia University of Bahawalpur, Pakistan.

---

## [Author Response · Author response to Decision Letter 0]

1 Jun 2021

response to reviewer #1, thank you for your contractive comments and suggestion, however please make clear the issues you want to raise, for eg in this manuscript your comment depends on biological replicates which is no more clear for author. thank you

---

## [Decision Letter · Decision Letter 1]

8 Jul 2021

PONE-D-20-38977R1

Tolerance to soil acidity of soybean (Glycine max L.) genotypes under field conditions at Southwestern Ethiopia

PLOS ONE

Dear Dr. Bedassa,

Thank you for submitting your manuscript to PLOS ONE. After careful consideration, we feel that it has merit but does not fully meet PLOS ONE’s publication criteria as it currently stands. Therefore, we invite you to submit a revised version of the manuscript that addresses the points raised during the review process.

ACADEMIC EDITOR: Although the revisions of the manuscript have been favorable, it is still necessary to make some modifications/improvements to the manuscript in order to be approved for publication in PLOS ONE.

Authors are requested to provide an adequate response to the reviewer's comments and submit a list of responses to the comments indicating specifically in which lines the improvements were done, or explaining why they were not considered. There are numerous grammatical and typo mistakes in the draft; therefore, language, wording and paraphrasing should be carefully reviewed and improved. The manuscript must be edited by a native English-speaking scientist or professional English editing service.

We look forward to receiving your revised manuscript.

Kind regards,

Saddam Hussain

Academic Editor

PLOS ONE

Journal Requirements:

Additional Editor Comments (if provided):

Although the revisions of the manuscript have been favorable, it is still necessary to make some modifications/improvements to the manuscript in order to be approved for publication in PLOS ONE.

Authors are requested to provide an adequate response to the reviewer's comments and submit a list of responses to the comments indicating specifically in which lines the improvements were done, or explaining why they were not considered. There are numerous grammatical and typo mistakes in the draft; therefore, language, wording and paraphrasing should be carefully reviewed and improved. The manuscript must be edited by a native English-speaking scientist or professional English editing service.

Reviewers' comments:

Reviewer's Responses to Questions

**Comments to the Author**

1. If the authors have adequately addressed your comments raised in a previous round of review and you feel that this manuscript is now acceptable for publication, you may indicate that here to bypass the “Comments to the Author” section, enter your conflict of interest statement in the “Confidential to Editor” section, and submit your "Accept" recommendation.

Reviewer #2: All comments have been addressed

Reviewer #3: (No Response)

Reviewer #4: (No Response)

Reviewer #5: (No Response)

2. Is the manuscript technically sound, and do the data support the conclusions?

Reviewer #2: Yes

Reviewer #3: No

Reviewer #4: Yes

Reviewer #5: Yes

3. Has the statistical analysis been performed appropriately and rigorously? 

Reviewer #2: Yes

Reviewer #3: N/A

Reviewer #4: Yes

Reviewer #5: Yes

4. Have the authors made all data underlying the findings in their manuscript fully available?

Reviewer #2: Yes

Reviewer #3: Yes

Reviewer #4: Yes

Reviewer #5: Yes

5. Is the manuscript presented in an intelligible fashion and written in standard English?

Reviewer #2: Yes

Reviewer #3: No

Reviewer #4: Yes

Reviewer #5: Yes

7. PLOS authors have the option to publish the peer review history of their article (what does this mean?). If published, this will include your full peer review and any attached files.

Reviewer #2: **Yes: **M. Asaad Bashir

Reviewer #3: No

Reviewer #4: **Yes: **Mohammad Reza Abdollahi

Reviewer #5: No

6. Review Comments to the Author

Reviewer #3: Manuscript has been poorly written and lacks coherence. Line N0.42 rewrite the sentence, sentence is long

Line. No 54 Grammatically incorrect, further the sentence doesn’t give any sence.

Line No 60- 63 rewrite the para.

Line No. 65 (Abush T) delete T.

Line No. 70 replace the word development by identification of .

Line No.71- 73 Sentence is too long and should be modified.

Introduction

It lacks coherence and the language is poor, which can be improved

Materials and methods

Line No. 84 Delete the sentence “According to the weather data obtained from the meteorological station of Mettu Agricultural Research Sub Center”.

Sampling and analysis.

Poorly written

It is not necessary to mention the whole procedure, only mention the method used and reference.

Line No. 125 replace amounts with amount.

Line No. 134 delete was.

Line No. 148 root volume.

Line No 149. Delete the sentence beginning from which-genotypes.

Line No. 151, Not clear, rewrite the sentence.

Results and discussion (sentences have been written long unnecessarily)

Line No. 184Replace the sentence as, Genotypes reflected significant differences for number of pods----------and above ground biomass in both limed and un limed soils.

Line No.190-191. The sentence doesn’t give any sense and should be deleted.

Reviewer #5: Comments

1. The abstract is well written and organized. line 17, “necessary” stage is not correct wording. I suggest to use more suitable word. Separate “BRS268had” into BRS268 had. Furthermore, abstract must be much more quantitative regarding the results and suggested innovation of the study.

2. The literature review needs to be more critical.

3. Please state clearly the novelty of the study and tested hypothesis at the end of the introduction.

4. Line 222: The highest HSW recorded was for PI423958 genotype not JMALM/PR142-15-SC genotype.

5. Table 6 is incomplete. Show HSW for lime and un limed or remove limed from the title if the limed treatments did not show any significant effect on HSW.

6. Provide reference to support the statement within line 235 to 239.

7. Throughout the manuscript, there are several grammatical errors and the English language used are poor or sub-standard.

8. The discussion section should be written well to clearly explain the science behind the results.

9. The paper fails to highlight the significant of the study and the conclusion needs to be expanded significantly to explain clearly about the novelty of the experimental setup. Also, policy recommendations can be suggested.

---

## [Author Response · Author response to Decision Letter 1]

2 Aug 2021

one reviewer is making the thing too big than making suggestion

---

## [Decision Letter · Decision Letter 2]

31 Aug 2021

PONE-D-20-38977R2

Tolerance to soil acidity of soybean (Glycine max L.) genotypes under field conditions at Southwestern Ethiopia

PLOS ONE

Dear Dr. Bedassa,

Thank you for submitting your manuscript to PLOS ONE. After careful consideration, we feel that it has merit but does not fully meet PLOS ONE’s publication criteria as it currently stands. Therefore, we invite you to submit a revised version of the manuscript that addresses the points raised during the review process.

ACADEMIC EDITOR: Although several opportunities have already been given to authors for improving the manuscript; the presentation of revised manuscript is still poor, and contains many typo and language errors that should have already been addressed by the authors. To avoid repeated rounds of revision it is likely that the manuscript will be rejected if significant improvements are not made in the returned draft.

Authors must revise the language of the draft by a professional English Editor/native speaker prior to submission. 

We look forward to receiving your revised manuscript.

Kind regards,

Saddam Hussain

Academic Editor

PLOS ONE

Journal Requirements:

Reviewers' comments:

Reviewer's Responses to Questions

**Comments to the Author**

1. If the authors have adequately addressed your comments raised in a previous round of review and you feel that this manuscript is now acceptable for publication, you may indicate that here to bypass the “Comments to the Author” section, enter your conflict of interest statement in the “Confidential to Editor” section, and submit your "Accept" recommendation.

Reviewer #3: (No Response)

Reviewer #5: All comments have been addressed

2. Is the manuscript technically sound, and do the data support the conclusions?

Reviewer #3: Partly

Reviewer #5: (No Response)

3. Has the statistical analysis been performed appropriately and rigorously? 

Reviewer #3: I Don't Know

Reviewer #5: (No Response)

4. Have the authors made all data underlying the findings in their manuscript fully available?

Reviewer #3: Yes

Reviewer #5: (No Response)

5. Is the manuscript presented in an intelligible fashion and written in standard English?

Reviewer #3: No

Reviewer #5: (No Response)

6. Review Comments to the Author

Reviewer #3: Abstract:

Abstract should be concise and precise especially results should be

Introduction:

Justification of the study is well grounded, though there is good scope for its improvement in light of the latest literature

Result and Discussion:

Discussion has been drafted monotonously, there is repetition of few sentences, which are self-explanatory and doesn’t need to be repeated again and again.

In the table genotype SCS-1 is reflecting lower yield under lime application, however genotype PI423958 is showing higher yield in comparison to SCS-1with lower yield attributes, I feel author needs to justify it.

Similarly, BRS268 genotype has reflected higher yield in un-limed soil in comparison to treated soil, which needs justification. Furthermore, yield attributes of the mentioned cultivar are better under treated soil, then how is it possible to yield more under untreated situation.

Line No.19 Insert which, between root volume and Where.

Line No.21 delete “and other’ insert (,)

Line No.22 delete (,) before and

Line No 23 delete “than un limed soils’

Rewrite the para from line No. 21 to Line No. 24, it is grammatically incorrect.

.

Minor corrections to be enacted.

Line No. 41 replace word acid by acidic and replace and by (,) between rainfall and leaching.

Line No. 42 crop management-------soil acidity Line No. 45 rewrite it.

Line No. 51 Delete (also) and (,), further more you are prolonging the sentence unnecessary, which doesn’t give any sense and reduces the quality of writing. So you better rewrite the sentence.

Line No.61 Modify the sentence, there is mention of the name of country twice in the same sentence.

Line No. 67 insert “Of ‘between are and primary.

Line No. 71 replace word tolerance with tolerant.

Line No. 72 identification of tolerant genotypes is not an amendment practice rather it is a management practice. Thus, requires modification.

Line No.87 delete “and’ and replace word “the’ with ‘an’

Line No. 89 is not clear , mention the mean values of both min. and max. temperature clearly.

Line No. 99 insert “of the soil samples was used to determine the bulk density’ after 105oC

Line No. 107 There is no need to write ‘that is among soil physical parameters’ so it should be deleted.

Line No.111

Line No. 134 Delete “was’ in the sentence. The treatments was comprised of two factors namely

Line No. 135 -137. No need to go for repetition of the sentences, I suggest for its deletion.

Line No. 151, Rewrite the sentence.

Minor corrections and typographical err

Line No. 187.Number of seed per plant.

Line No.190 insert, all yield components and yield. And replace were with word ‘was’

Line No. 192.delete toxicity.

Line No. 193 Insert pods and seeds instead of pod and seed per plant.

Line No. 194.word that in limed to be replaced by treated soil.

Line No. 198 word material to be replaced by genotype.

Line No. 204- 206 percentage increase or decrease against treatment is ambiguous and needs clarity.

Line no 223-225 revisit the results, and interpret them clearly as the tables mentioned by author.

Line No. 225 insert “response” after variable.

Line No. 226 doesn’t to be replaced by didn’t.

Line No. 227.Rewrite the Justification given by the author to variable response of genotypes under lime application to 100 seed weight.

Line No. 236 is self-contradictory, elucidate on it.

Line No. 290

Line No. 292 write number of nodules instead of nodule.

Line No. 292-295, rewrite the sentence.

Line No.304-05 check the justification given by Kuswantoro.

Line No. 307. root length and root hair (what is the difference).

Line No.318-320. Justification is not in accordance with the results.

Line No. 320 -325 Justification given by author is entirely different, replace it or come with some clear support.

Line No. 404-405. Give some concreate justification or support.

Table No. 6 under limed and un-limed soils, where are the limed and un-limed columns.

The author has mentioned that the experiment was conducted during 2017/18, where as in the tables he highlights one year data. The author should come clear on it.

Conclusion:

Re write the conclusion, there is no need to elaborate the results in conclusion section. conclusion should be short and precise.

The manuscript needs thorough revision for language editing by a professional native English speaker.

This manuscript also requires a technical review done critically by senior author and remove all the typos present throughout the manuscript.

Reviewer #5: (No Response)

7. PLOS authors have the option to publish the peer review history of their article (what does this mean?). If published, this will include your full peer review and any attached files.

Reviewer #3: No

Reviewer #5: **Yes: **Emmanuel Abban-Baidoo

---

## [Author Response · Author response to Decision Letter 2]

11 Oct 2021

Please both reviewer see carefully what i have incorporated based on your comments in the body of the paper , i have touched all your comments and questions thank you for your time and efforts

---

## [Decision Letter · Decision Letter 3]

21 Dec 2021

PONE-D-20-38977R3Tolerance to soil acidity of soybean (Glycine max L.) genotypes under field conditions at Southwestern EthiopiaPLOS ONE

Dear Dr. Bedassa,

Thank you for submitting your manuscript to PLOS ONE. After careful consideration, we feel that it has merit but does not fully meet PLOS ONE’s publication criteria as it currently stands. Therefore, we invite you to submit a revised version of the manuscript that addresses the points raised during the review process.

Although the revisions of your manuscript have been favorable. I have personally checked the revised manuscript and found that it still requires several corrections/improvements. Please note that this is the final chance of revisions. Provide an adequate response to each of the comment, and submit a list of responses to the comments indicating specifically in which lines the improvements were done, or explaining why they were not considered.

In addition to the reviewer’s comments; address the following issues;

Please strictly follow the journal’s guidelines for whole draft. Specifically, the abstract is not as per suggested format of journal. You may follow some recently published articles in PLOS One.  

Language needs to be carefully checked prior to submission of revised manuscript. There are still various grammatical and typo mistakes throughout the manuscript.

Be consistent regarding treatment description and the usage of units/abbreviations. All the abbreviations should be defined at first mentioned place.

All the tables should be self-explanatory. Check the uniformity of all tables.

In introduction, add a statement to highlight the research gap/novelty of the study prior to objectives.

Avoid starting a sentence with number/abbreviation.

Table 10: Decrease percentage? Compared with what?

We look forward to receiving your revised manuscript.

Kind regards,

Saddam Hussain

Academic Editor

PLOS ONE

Journal Requirements:

Additional Editor Comments (if provided):

Editor's comments:

Although the revisions of your manuscript have been favorable. I have personally checked the revised manuscript and found that it still requires several corrections/improvements. Please note that this is the final chance of revisions. Provide an adequate response to each of the comment, and submit a list of responses to the comments indicating specifically in which lines the improvements were done, or explaining why they were not considered.

In addition to the reviewer’s comments; address the following issues;

Please strictly follow the journal’s guidelines for whole draft. Specifically, the abstract is not as per suggested format of journal. You may follow some recently published articles in PLOS One.

Language needs to be carefully checked prior to submission of revised manuscript. There are still various grammatical and typo mistakes throughout the manuscript.

Be consistent regarding treatment description and the usage of units/abbreviations. All the abbreviations should be defined at first mentioned place.

All the tables should be self-explanatory. Check the uniformity of all tables.

In introduction, add a statement to highlight the research gap/novelty of the study prior to objectives.

Avoid starting a sentence with number/abbreviation.

Table 10: Decrease percentage? Compared with what?

Reviewers' comments:

Reviewer's Responses to Questions

**Comments to the Author**

1. If the authors have adequately addressed your comments raised in a previous round of review and you feel that this manuscript is now acceptable for publication, you may indicate that here to bypass the “Comments to the Author” section, enter your conflict of interest statement in the “Confidential to Editor” section, and submit your "Accept" recommendation.

Reviewer #6: All comments have been addressed

Reviewer #7: (No Response)

2. Is the manuscript technically sound, and do the data support the conclusions?

Reviewer #6: Yes

Reviewer #7: (No Response)

3. Has the statistical analysis been performed appropriately and rigorously? 

Reviewer #6: Yes

Reviewer #7: (No Response)

4. Have the authors made all data underlying the findings in their manuscript fully available?

Reviewer #6: Yes

Reviewer #7: (No Response)

5. Is the manuscript presented in an intelligible fashion and written in standard English?

Reviewer #6: Yes

Reviewer #7: (No Response)

6. Review Comments to the Author

Reviewer #6: I think the authors addressed all the reviewers comments and I recommended to accept the manuscript for publication

Reviewer #7: - The subject addressed is within the scope of the journal.

- The research contributions of the paper should be articulated more clearly. The abstract is not representative of the content and contributions of the paper. The abstract does not seem to properly convey the rigor of research.

- Aside from the aim stated in the title, the research gap and the goals of the research are not specified which leads to the reader missing the significance of the research.

- However, the manuscript, in its present form, contains several weaknesses. Appropriate revisions to the following points should be undertaken in order to justify recommendation for publication.

- For readers to quickly catch your contribution, it would be better to highlight major difficulties and challenges, and your original achievements to overcome them, in a clearer way in abstract and introduction.

- It is suggested to add articles entitled “Abderrahmane et al. Influence of Highway Traffic on Contamination of Roadside Soil with Heavy Metals”, “Nkansah et al. Preliminary Studies on the Use of Sawdust and Peanut Shell Powder as Adsorbents for Phosphorus Removal from Water” and “J. Sam. Compressive Strength of Concrete using Fly Ash and Rice Husk Ash: A Review” to the literature review.

- This raises some concerns regarding the potential overlap with authors previous works. The authors should explicitly state the novel contribution of this work, the similarities and the differences of this work with their previous publications.

- Some key parameters are not mentioned. The rationale on the choice of the particular set of parameters should be explained with more details. Have the authors experimented with other sets of values? What are the sensitivities of these parameters on the results?

- Please avoid reference overkill/run-on, i.e. do not use more than 3 references per sentence.

- Some assumptions are stated in various sections. Justifications should be provided on these assumptions. Evaluation on how they will affect the results should be made.

7. PLOS authors have the option to publish the peer review history of their article (what does this mean?). If published, this will include your full peer review and any attached files.

Reviewer #6: No

Reviewer #7: No

---

## [Author Response · Author response to Decision Letter 3]

20 Apr 2022

we acknowledge all reviewers for their constructive ideas and comments, however some reviewers are making complex than touching the target problems

---

## [Editor Report · Decision Letter 4]

29 Jul 2022

Tolerance to soil acidity of soybean (Glycine max L.) genotypes under field conditions at Southwestern Ethiopia

PONE-D-20-38977R4

Dear Dr. Bedassa,

We’re pleased to inform you that your manuscript has been judged scientifically suitable for publication and will be formally accepted for publication once it meets all outstanding technical requirements.

Kind regards,

Saddam Hussain

Academic Editor

PLOS ONE
---

## [Editor Report · Acceptance letter]

18 Aug 2022

PONE-D-20-38977R4 

Tolerance to soil acidity of soybean (*Glycine max L.*) genotypes under field conditions atSouthwestern Ethiopia 

Dear Dr. Bedassa:

I'm pleased to inform you that your manuscript has been deemed suitable for publication in PLOS ONE. Congratulations! Your manuscript is now with our production department. 

Kind regards, 

on behalf of

Dr. Saddam Hussain 

Academic Editor

PLOS ONE